# Gas Dynamics Processes above the Polymers Surface under Irradiation with Broadband High-Brightness Radiation in the Vacuum Ultraviolet Spectrum Region

**DOI:** 10.3390/polym14193940

**Published:** 2022-09-21

**Authors:** Aleksei Pavlov, Tadeush Shchepanyuk, Andrei Skriabin, Victor Telekh

**Affiliations:** Educational and Scientific Center of Photonic Energy, Bauman Moscow State Technical University, 105005 Moscow, Russia

**Keywords:** PTFE, polymer ablation, VUV radiation, plasma flows, extreme environment

## Abstract

Obtaining new data on the gas-dynamic responses from the polymer samples (polytetrafluoroethylene, PTFE) irradiated by powerful VUV radiation from compressed plasma flows is in the focus of the present study. An erosion type magnetoplasma compressor (MPC), a type of plasma focus discharge, was used as a radiation source. The operating voltages of the MPC were between 15 and 25 kV, the maximum measured discharge current was 200 kA, and the radiation energy in the VUV range was ≈1–2 kJ. The VUV fluxes on the sample surface were high and equal to ≈10^22^–10^24^ photons cm^−2^·s^−1^. Double-exposure laser holographic interferometry and schlieren photography were used to diagnose and visualize the gas-dynamic structures. The spatial distribution of the parameters (temperature, pressure and concentrations of electrons and ions) was defined based on the assumption of local thermodynamic equilibrium. It has been demonstrated that the maximum temperature ranged from ≈ 10 to 15 kK in the plasma layer. The electron concentration was ≈ (0.7–1.6) × 10^18^ cm^−3^ in this region. The used techniques of optical diagnostics and procedures of result processing make it possible to obtain data on the dynamics of polymer ablation, which occurs when their surface is exposed to powerful energy fluxes (thermal, shock-wave, radiation, and other extreme loads).

## 1. Introduction

Polymers (PTFE, POM and others) are considered as propellants for electric propulsion systems (EPS), such as laser thrusters and pulsed plasma thrusters (PPT), where an impulse is generated due to polymer ablation under extreme heat and radiation loads. The main plasma generating processes which determine the propellant flow rate are laser ablation (in laser thrusters) and light erosion (in PPT) [1,2,3]. Nowadays, these EPS active development takes place alongside nano- and pico-satellites [4,5]. Unlike gases, solid polymer propellants have advantages in PPT miniaturization, precise parameters control, operation simplification, etc.

The term “ablation” means elimination of an upper layer from the surface of a solid body under extreme environments, which include a lot of energy impacts such as high temperature and mechanical loads, electromagnetic radiation, charged particle and atomic cluster fluxes, and other phenomena. In case of laser and pulse plasma thrusters an extreme radiation fluxes are the main factor, which causes ablation and thrust appearance. Unlike laser propulsion systems, light-induced ablation (or light erosion) occurs in PPT under exposure of a high-brightness broadband radiation in the UV-VUV ranges.

Despite the development and research long history of PPTs (since the 1950s) [6,7] and laser propulsion (since the 1970s) [8], the problem of describing both the processes of light erosion and laser ablation by powerful radiation (with a radiation flux more than 10^6^ W·cm^−2^) remains. Light induced ablation of polymers is characterized by a lot of approaches to describe this process (for example, [9,10,11]), but these models are based on studies of laser radiation exposure to the surface. There are also a significant number of studies about low-intensity UV/VUV radiation fluxes’ exposure on polymers [12,13,14,15,16,17,18], which does not lead to noticeable ablation and is aimed at modifying or degrading their surfaces. In the case of PPT, the radiation impact on the surface is much more extreme (the photon flux density is about 10^24^ photons·cm^−2^·s^−1^), which stimulates the appearance of ablation flows. At the same time, data on polymer ablation under such extreme environment are practically absent in the literature. Besides, such studies are further complicated by the complexity of the radiation sources [19,20,21,22].

To explain some features of physical and chemical processes during irradiation of polymers by the high-brightness VUV / UV fluxes, it is necessary to use special diagnostic techniques. These techniques will allow us to establish the features of the gas-dynamic “response” from the irradiated surface and plasma parameters, which determine the PPT thrust and lifespan characteristics. The purpose of this study is to obtain new information about light-induced ablation of polymers under the high-power broadband VUV radiation.

Note that, in the case of laser ablation, the exposure of focused radiation results in crater formation on the surface. However, in many practical applications, such as PPT, radiation- and heat-resistant materials and others, extreme energy fluxes are loaded to the entire material surface. The same effect is realized in our experiments.

## 2. Experimental Methods

### 2.1. Key Features of Used Plasma Radiation Source and its Operation Process

In this work, a promising source of the high-brightness VUV radiation was used. This is a short-wave non-synchrotron emitter based on a coaxial high-current plasma accelerator. This accelerator is a magneto plasma compressor (MPC) of the erosion type [20,23]. In the MPC, plasma-dynamic heating of the plasma is carried out as a result of shock-wave thermalization of a directed kinetic energy of high-speed plasma flow during its deceleration in a background gas. In this case, a high-temperature emitting region (a plasma focus) is formed on the axis near the central electrode [23]. Such discharges are characterized by high spectral-brightness characteristics of radiation in the UV and VUV ranges and have an extended glow region [21,24].

For discharges in gases, the background gas acts as a “cutter-off” for the hard component of the plasma emission spectrum. The short-wavelength limit of the emission spectrum of such discharges is determined [25] by the first ionization potential of the background gas (for inert gases) or by the Schumann-Runge bands (for oxygen containing media). As a consequence, as the intensity rise time *τ_r_* of the radiation is determined only by temporal dynamics of the plasma flow energy in the deceleration zone, there is a significant mismatch in the *τ_r_* values and the time constants *τ_c_* of the electrical circuit. This makes it possible to generate powerful VUV radiation pulses with a steep leading edge.

Plasmodynamic, radiative, and electrical parameters of this MPC have been presented in a number of articles and reviews [20,21,23,24]. These parameters determine the discharge features (with a conversion efficiency of ≈70–90%) and radiation emission (with an emission efficiency of ≈40–60% off the plasma energy). Unlike other short-wave radiation sources [12,17], the VUV radiation fluxes range from 10^20^ to 10^24^ photons cm^−2^·s^−1^ in the near zone of the MPC.

### 2.2. Experimental Set up

An erosion-type MPC, with coaxial electrodes (with diameters of 6 and 34 mm) and a PTFE dielectric hub, was installed in a vacuum chamber with a volume of 1.5 × 10^−2^ m^3^ (see Figure 1). This chamber was pump down before each discharge to a pressure of *p* ≤ 1 Pa and then it was filled with a gas up to *p* = 10^3^−10^4^ Pa. A low-inductance capacitor (with a capacity of 18 μF and an operating voltage of 25 kV) was switched to the MPC via a thyratron (Pulsed Technology Ltd., Ryazan, Russia). Current waveforms were recorded with Pearson current monitor 110 (Pearson Electronics, Palo Alto, CA, USA) and Tektronix TDS 2024b oscilloscope (Tektronix, Beaverton, OR, USA).

PTFE bars (with dimensions of 30 mm × 50 mm and a thickness of 10 mm) were used as samples for the VUV exposure. They were installed with their long side along the discharge axis at a distance of 45 mm from the MPC axis. Thus, the nearest sample end to the MPC barrel received 2–2.5 times more energy than the far end. This made it possible, among other things, to register different modes of expansion of the ablation products [26].

### 2.3. Laser Diagnostics Techniques

Optical laser diagnostics was used for visualization and quantitative study of a gas-dynamics response from the PTFE samples under the VUV irradiation. It included the double-exposure holographic interferometry and the bright field schlieren imaging [26,27,28,29].

The laser holographic interferometry has been described in detail in a previous study [26] and its scheme is presented in Figure 2. The diagnostics was implemented with a Nd:YAG Solar LQ-115 (Solar laser systems, Minsk, Belarus) laser 1 (532 nm). In the interferometer scheme, a laser beam was splitted into probing and reference one in the intensity ratio of 1:1 by a splitter 2. The final intensity ratio in the arms was selected experimentally using a neutral-density filter 3 located in front of the hologram. The discharge radiation was cut off by a narrow-band interference filter 4 (λ = 532 nm, Δλ = 10 nm). The lengths of the optical paths with an accuracy of 1 cm were aligned by a mirror *5* mounted on a graduated rail. An aperture of the studied objects was up to 200 mm in diameter.

The probing beam was expanded by a laser beam expander 8 and lenses 9 and 10. After passing through the phase inhomogeneity in a vacuum chamber 18, the beam was collected on a screen 19 by a system of lenses 11–15. The mutual arrangement of lenses 12–15 made it possible to obtain a clearly focused image of an object with an almost compensated level of distortion. The reference beam was reflected from mirrors 6 and 7, which aligned the optical paths, expanded with telescope from lenses 16, 17 and got into the hologram. To record finite fringe interferograms, a quartz wedge 20 with an apex angle of 5° was introduced into the probing beam. The wedge was rotated around the vertical axis by an angle of 3–5° between two exposures and it changed the angle between the reference and probing beams. The distance between the interference fringes at these angles was ≈1–0.5 mm. The hologram [28] was captured on a VRP-M holographic film (Slavich Ltd., Pereslavl-Zalessky, Russia) sensitive to the green wavelength range. At the same time, each point of the hologram corresponded to a certain point of the object. This made it possible to reconstruct images from the holograms with a Sony Alpha DSLR-A700 camera by placing the film approximately at an exposure angle in the white light of an ordinary presentation projector.

To capture schlieren images, the reference beam was blocked by a screen 21, and a diaphragm 22 with a diameter of 1.2 mm was placed at the focus of a lens 12 with a focus length of 400 mm. Both the VRP-M film and a white matte screen were installed in place of a screen 19, which allowed using of a CCD Videoscan VS-285C camera 23 (Videoscan Ltd., Moscow, Russia) for quick selection of diagnostic parameters. A moment of laser triggering was detected by a signal from a photodiode 24.

The maximum spatial resolution of the CCD camera was 78 line pairs per mm (for the camera) and 1570 line pairs per mm (for the film). The spatial resolution of the diagnostic unit was determined by the resolution of the optical system and the registration system (from 100 μm for the digital camera and less than 50 μm for the light-sensitive film), and the temporal resolution (10 ns) was determined by the laser pulse duration. So, the used technique allowed us to visualize large optical fields (in our case 150 mm) with a minimal spatial resolution of 50 microns at any point of the frame. Synchronization of the laser pulse and photo capturing was fulfilled with a BNC 575 Digital Generator *26* (Berkeley Nucleonics Corporation, San Rafael, CA, USA).

### 2.4. To Analysis of Holograms

The used laser diagnostic methods allow us to capture holograms and schlieren images of the gas-dynamic response above the irradiated PTFE surface at different points in time. The refractive index *n*(*x*, *y*, *z*) of a medium through which the probing laser beam passes is obtained by processing of holograms (interferograms). On the other hand, data on a gradient of *n*(*x*, *y*, *z*) (and correspondingly a density gradient) can be estimated by processing the schlieren images.

As a result of disturbances, the refractive index *n* can be written as [30]:*n*(*x*, *y*, *z*) = *n*_0_ + Δ*n*(*x*, *y*, *z*).(1)

Here, *n*_0_ is a refractive index of the undisturbed media and Δ*n*(*x*, *y*, *z*) is its variance due to disturbances.

The scheme of the probing beam passing through the inhomogeneity is shown in Figure 3. The scan direction corresponds to the *z*-axis. The difference between the coordinates *z*_1_ and *z*_2_ is equal to a thickness of the studied object as *L* = *z*_2_ − *z*_1_. Here, *z*_1_ and *z*_2_ are the entry and exit coordinates of the beam into the inhomogeneity. So, we can assume that the optical path length of the probing laser beam is constant along the studied surface (*x*-axis). Due to the extended dimensions of the surface, it is possible to neglect optical effects at the sample edges. As a result, it is possible to obtain data on the 2D fields of the refractive index *n*(*x*, *y*) and other parameters.

In view of the above, the insertion of the optical inhomogeneity into the probing arm of interferometer causes a phase shift Δ*φ*(*x**, y*) of the recorded wave as
(2)$$\Delta \varphi \left(x,y\right)=\frac{2\pi }{\lambda }\int_{z_{1}}^{z_{2}}\Delta n\left(x,y,z\right)dz.$$

Here, *λ* = 532 nm is the probing wavelength. Equation (2) can be integrated along the beam path inside the inhomogeneity. The fringes shift at some point of the interferogram can be determined as
(3)$$\Delta m=m\left(x,y\right)-m_{0}=\frac{\Delta \varphi \left(x,y\right)}{2\pi }.$$

Here, *m*_0_ is an order of the interference fringe in the absence of inhomogeneity, *m*(*x, y*) is a fringe order at the same point in the inhomogeneity presence. With the geometric optics approximation at small angles of deflection, Equation (2) can be reduced to [30]
(4)$$\Delta \varphi \left(x,y\right)=2\pi L\Delta n\left(x,y\right)/\lambda .$$

So, it is possible to find the variance of Δ*n*(*x*, *y*) with values of the fringe shifts Δ*m*(*x*, *y*) as
(5)$$\Delta n\left(x,y\right)=\Delta m\left(x,y\right)\lambda /L.$$

With data on Δ*n*(*x*, *y*), the refractive index can be calculated as
(6)$$n\left(x,y\right)=\Delta m\left(x,y\right)\lambda /L+n_{0}.$$

Thus, it is possible to find the values of the integral refractive index by determining from the hologram Δ*m*(*x*, *y*). A hypothetical trend of the interference fringes paths in different situations is presented in Figure 4. To obtain absolute values of *n*(*x*, *y*), it is necessary to count it from the refractive index in vacuum (see the line *VV*).

Since the vacuum chamber is filled with an inert gas with an initial concentration of *N*_0_, the fringes are shifted by a value of Δ*m**_V_* relative to the observed vacuum fringes as
(7)$$\Delta m_{V}=\frac{2\pi \alpha_{G}N_{0}L}{\lambda }$$

Here, *N*_0_ is the background gas concentration, $\alpha_{G}$ is the background gas polarizability [31]. At a shock wave front a sharp increasing of a concentration and a pressure shifted by a value of Δ*m_m_*. In this case an absolute shift Δ*m_abs_* is equal to Δ*m_abs_* = Δ*m_m_* + Δ*m_V_*. Here, a value of Δ*m_m_* is positive.

On other hand, neutral atoms, molecules, ions (including molecular ones) and electrons can be present in studying gas-plasma system. A contribution of various particles to the refractive index can be considered additive as [29]
(8)$$n-1=\sum_{k}C_{k}N_{k}$$

Here *C_k_* is a refraction of the *k*-th particle per a one particle and *N**_k_* is a concentration of the *k*-th particle. The refractive index in the visible range is described by the Cauchy’s equation [30]:(9)$$C_{a}N_{a}=\left(A+B/\lambda^{2}\right)\cdot N_{a}/N_{L}$$

Here *N**_L_* is the Loschmidt’s number, *N_a_* is the concentration of atoms, and *A* and *B* are constants for a gas. Usually *B**/λ*^2^ << *A*, and Equation (9) can be written as
(10)$$C_{a}N_{a}=2\pi \sum_{x}\alpha_{x}N_{x}$$

Here, *N_x_* is a fraction of the *x*-th sort of atoms or ions in the plasma and *α*_a_ is the dipolar polarizability of an atom or ion [31]. An electron contribution to the plasma refraction is given by [32]:(11)$$C_{e}N_{e}=-\varphi \lambda^{2}N_{e}$$

Here, *φ* = 4.49 × 10^−14^ cm and *N_e_* is the electron concentration. So, the contributions for the electronic and ionic (atomic) components give opposite signs. In the present study, we used the equation for relations between the plasma refractive index and its composition as
(12)$$n-1=2\pi \sum_{x}\alpha_{x}N_{x}-\varphi \lambda^{2}N_{e}$$

Analysis of Equation (12) allows us to relate a negative shift of Δ*m_m_* to a significant influence of free electrons due to an effective ionization in the plasma. According to Equation (12), the refractive index is also determined by the fields of temperature T and pressure *p* as *n* = *n*(*T*, *p*). A physical model taken for the numerical modeling of the plasma parameters with the measured data on the refractive indexes can be briefly formulated as follows. The plasma is considered as a system in local thermodynamic equilibrium (LTE). In this case all kinetics processes can be neglected, i.e., their relaxation time is supposed to be much shorter than characteristic time of the plasma existence and evolution, and all its local properties are determined by two parameters, i.e., temperature *T* and pressure *p* (or density *ρ*). The PTFE plasma composition was calculated with the ionization equilibrium model for weakly non-ideal plasma based on the Saha-Eggert equation. The interaction of charged particles was taken into account using corrections in the Debye approximation for the grand canonical ensemble. The calculation technique has been described in detail in [33].

## 3. Results and Discussion

### 3.1. Experimental Data on Gas-Dynamic Response from PTFE Surface

With the recorded current waveform, it was found that the energy input to the discharge was ≈3.6 kJ, with ≈30% during the first half period. Maximum current was up to ≈150 kA, and half period of the discharge was ≈6 μs (see Figure 5). Taking into account discharge and light emission efficiencies, the total radiation energy from the discharge reached ≈1 kJ. The radiation flux in the visible spectral range at the end of the PTFE bar close to the MPC maximally was estimated at ≈10^5^ W·cm^−2^. It was estimated at ≈10^6^ W·cm^−2^ for the VUV range (here it was assumed that the first half period accounted for 2/3 of the energy). The moment of the image capturing corresponded to a stage of the MPC discharge (with a time of ≈ 11 μs from the thyratron switching) with a minimum of shock impact on PTFE. This was detected as a sharp peak of a photodiode signal.

A special feature of the MPC radiation generation is the fact that the discharge emitted in the visible range during ≈3 μs. This caused the PTFE ablation with the formation of C and F vapors. When a plasma focus has occurred (after ≈3 μs), the discharge spectrum was shifted to the VUV range with a photon energy of ≈15–20 eV. This circumstance led to efficient ionization of the C and F vapors with ionization potentials of *I**_C_* = 11.3 eV and *I**_F_* = 17.4 eV. Ionization of the background gas did not occur due to its high ionization energy (*I**_Ne_* = 21.6 eV). Figure 6 shows schlieren images of processes in discharge zones and gas-dynamic response above the irradiated surfaces for discharges in neon and neon/air mixture.

It was shown that the gas-dynamic response in the case of discharge in pure neon significantly differed from discharge in the neon/air mixture. In the case of hard photons (discharge in neon), radiation of the visible range heated the PTFE surface and generated C and F vapors due to ablation. Further photoionization by hard VUV photons caused an emergence of the plasma layer that created a “plasma piston” when heated and expanded. The moving piston compressed the background gas and generated a shock wave 1 (see Figure 6). Therefore, two zones were distinguished above the irradiated samples: the shock-compressed gas (separated from the background gas by the shock wave front 1) and the PTFE plasma between a contact boundary 2 and the irradiated surface 3. It was found that the sample ablation occurred without a mechanical impact from the shock wave front 4 from the MPC. In the case of soft photons (discharge in the neon/air mixture), the visualized response (the shock wave front 1 and the contact boundary 2) was less pronounced. Another important difference was a presence of an optical thick layer 5 near the surface 3 irradiated in neon. Its structure was, in our opinion, formed by a cloud of dense vapors of the ablative PTFE bars.

A comprehensive analysis of the captured schlieren images and interferograms makes it possible to determine parameters of the shock wave, the shock-compressed gas, and the photon-induced plasma layer.

Figure 7 presents interferograms of the gas-dynamic response from the irradiated PTFE surface. Five interference fringes (#10, 20, 30, 40 and 50) were numbered and selected for further consideration. These fringes differed by a distance *l* from the MPC barrel: e.g., *l* = 41.5 mm (for fringe # 10), and *l* = 17 mm (for fringe #50).

For the 50-th fringe (see Figure 7b), the background gas compression was approximately equal to *ρ*_1_/*ρ*_0_ ≈ 1.8. This value was resulted from a comparison between positions of the shock wave front and the contact boundary in the Figure 7b. Here and further on, the subscript «0» corresponds to the background gas parameters, and the subscript «1» corresponds to the parameters behind the shock front. The fact that the background gas density at the shock front has changed by a factor of 1.88 can also be determined with data on interference shifts.

Using the shock adiabat, the pressure ratio *p*_1_/*p*_0_ can be determined with data on the ratio *ρ*_0_/*ρ*_1_ as [34]:(13)$$\frac{p_{1}}{p_{0}}=\left(\frac{\gamma +1}{\gamma -1}-\frac{\rho_{0}}{\rho_{1}}\right){\left(\frac{\gamma +1}{\gamma -1}\cdot \frac{\rho_{0}}{\rho_{1}}-1\right)}^{-1}$$

Under the implemented conditions for a monatomic gas (with an adiabatic index of *γ* = 1.67) for fringe #50 the pressure ratio was *p*_1_/*p*_0_ ≈ 3.1, and the absolute pressure behind the shock wave front was *p*_1_ ≈ 1.64 × 10^5^ Pa. With distance from the source, the fraction of radiation flux on the target decreased. This led to lower velocities of the contact boundary and consequently to lower pressures behind the shock wave front. Therefore, for fringe #20, the pressure ratio was *p*_1_/*p*_0_ ≈ 2.14 at the absolute pressure of *p*_1_ = 1.13 × 10^5^ Pa. It can be seen that, after a jump at the shock wave front, the interference fringes were changed, practically without a slope to the contact boundary. This means that at the time of the image capturing, the pressure distribution in the shock-compressed gas was practically uniform. Therefore, the pressure at the contact boundary was equal to the pressure behind the shock front (see points C and D in Figure 7d). Based on the estimates for the energy ratios [34], the temperature *T*_1_ of the shock-compressed gas was minor and the overheating did not exceed *T*_1_−*T*_0_ ≈ 100–150 K.

### 3.2. Interpretation of Gas Flow Structures with Interferograms

With an increase in the gas density, the fringes shifted to the right relativity the “vacuum line” (*VV*). The shift magnitude was characterized by the value of Δ*m**_V_* (see Equation (7)). An interpretation of the medium parameter changing from a non-disturbed state (far away from PTFE) to the PTFE sample is presented below.

The fringe #20 shape is shown in Figure 7d. The AB segment corresponded to an undisturbed background gas. The segment BC was interpreted as a density jump at the shock wave front. The CD curve in this figure corresponded to shock-compressed background gas. The contact boundary (acting as a moving “plasma piston”) between the shock-compressed gas and the vapor plasma was visualized as the segment DE. The section DE limited the region of finite dimensions with the temperature field rearranging from *T*_1_ (in the shock-compressed gas region) to *T_pl_* ≈ 7–20 kK (in the plasma region). As applied to the electric spark discharge, such regions are called the «shell» [35]. The EFGH curve was demonstrated the *n*(*x*, *y*) changing in the plasma and represented the complex dependence of its thermodynamic parameters in the height above the PTFE surface.

### 3.3. Thermodynamic Parameters of Plasma above PTFE Surface

Figure 8 presents isolines of the total atom and ion concentrations $N_{g}=\sum_{x}N_{x}$ (solid curves) and the refractive indexes (*n*(*x*, *y*) − 1) (dashed curves) in the *p*-*T* coordinates constructed in the LTE approximation with the mentioned above calculation technique. Obviously, the (*n*(*x*, *y*) − 1) values characterize the optical properties of the plasma and they are uniquely associated with p and T with the LTE assumption. As noted above, it is possible to evaluate the refractive indexes (*n*(*x*, *y*) − 1) with data on the fringes shifts. By determining the pressure field *p*(*x*, *y*) in some way, the temperature field *T*(*x*, *y*) was determined. Consequently, other thermodynamic parameters, such as enthalpy, ionization degree, or sound speed, can be estimated too.

The fringes shift at each point of the curve CDEFG was determined with the interferograms (see Figure 7) in the terms of the Δ*m_abs_* absolute values, i.e., relative to the vacuum fringe *VV*. The points were indicated for fringe #20 in Figure 7d. For fringes #10, #30, #40 and #50 the point designation was the same. Equation (6) was used for calculation of the refractive index (*n*(*x*, *y*) − 1) with the measured fringe shift Δ*m_m_*(*x*, *y*).

The pressure distribution in the plasma layer above the irradiated sample was estimated with the obtained data on *n*(*x*, *y*). Assuming that the contact boundary had a velocity lower than the local sound velocity, it can be assumed that the pressure in the high-temperature region was practically constant [35]. For further reasoning, we assume that the pressure at the surface (point G) increased by no more than 10% compared with point E. The boundary layer between the plasma and the PTFE sample (region GH) in this study was visualized as an extremely thin region near the surface, but it has not been reliably resolved. Further estimation of the plasma parameters is presented below for fringe #20. The pressure has been *p*_1_ = 1.13 × 10^5^ Pa (for point E at the contact boundary) as seen from above. The absolute fringe shift was Δ*m_abs_* = 2.3 that corresponded to the value of (*n*(*x*, *y*) − 1) = 4.1 × 10^−5^. We found point E in Figure 8 with these data. The plasma temperature was about ≈6 kK at this point. Point F was at the intersection of the line *p* = p_1_ with the curve (*n*(*x*, *y*) − 1) = 0.89 × 10^−5^. Point G lied at the intersection of the line *p* = 1.1∙*p*_1_ = 1.24 × 10^5^ Pa and the curve (*n*(*x*, *y*) − 1) = 3.5 × 10^−5^. Thus, curve EFG (Figure 7d) in the *p*-*T* coordinates was displayed by the curve EFG(2) (see Figure 8). Similarly, we constructed the curves EFG for the fringes #10, #30, #40 and #50. Thus, the plasma temperature varied (for fringe 20) from ≈6 kK at the contact boundary to ≈13 kK inside the plasma layer and dropped to ≈7 kK (to the boundary layer at the sample surface). Electron concentration in the plasma varied in the range of ≈(0.7–1.4) × 10^18^ cm^−3^. Getting closer to the radiation source, plasma parameters changed significantly. For the #50 fringe the PTFE plasma temperature increased from ≈ 8 kK at the contact boundary to ≈ 15 kK inside the plasma layer and then decreased. Electron concentration in the plasma has changed in the range of ≈(0.7–1.6) × 10^18^ cm^−3^. The distributions of the maximum plasma temperature along the sample length are shown in Figure 9.

## 4. Conclusions

This study discusses the radiation-induced plasmodynamic structures diagnostics and their quantitative description technique. These structures can be generated due to the exposure of the broadband (from visible to VUV range) high-power (with a radiation flux up to of ≈ 10^6^ W·cm^−2^) radiation to polymer samples (PTFE). With the suggested techniques, it is possible to determine both the parameters of the generated gas flows (pressure, density in the shock-compressed layer, etc.) and the thermodynamic parameters inside the radiation-induced plasma layer (pressure, temperature, and concentrations). In the LTE approximation, it was demonstrated that the maximal temperature reached ≈ 10 to 15 kK in the plasma layer. The electron concentration was ≈ (0.7–1.6) × 10^18^ cm^−3^ in this region. The obtained data can be useful to clarify of some workflow features of the spacecraft plasma thrusters with solid polymer propellants.

## Figures and Tables

**Figure 1 polymers-14-03940-f001:**
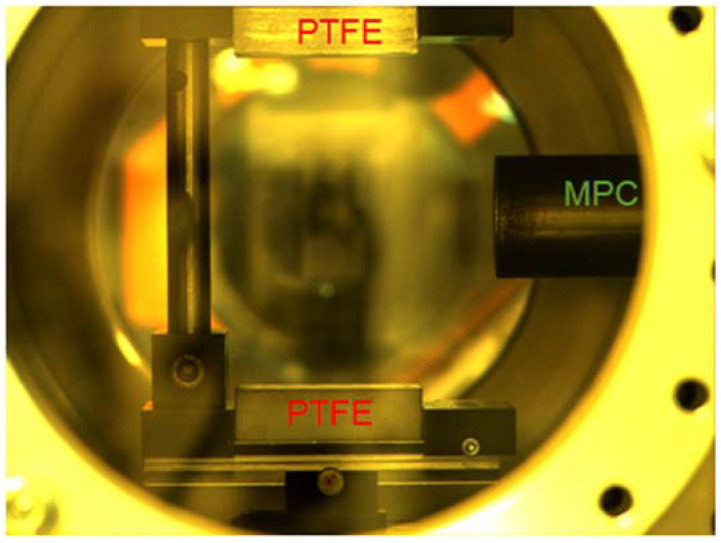
Photo of PTFE bars in experimental set up.

**Figure 2 polymers-14-03940-f002:**
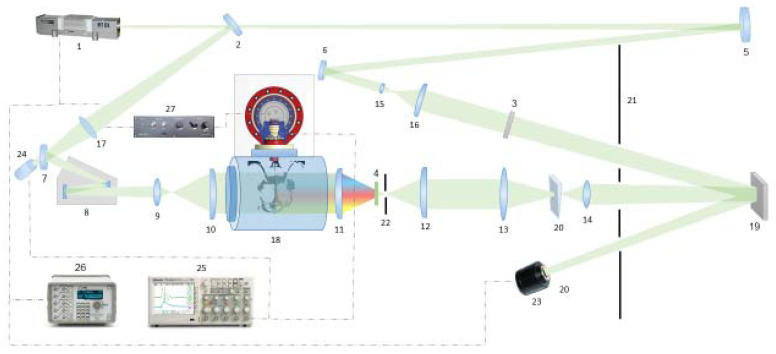
Scheme of optical diagnostics: 1—Solar Nd:YAG LQ115 laser (532 nm), 2—splitter, 3—neutral-density filter, 4—interference filter (λ = 532 nm, Δλ = 10 nm), 5, 6, 7—mirrors, 8—laser beam expander, 9—17—lenses, 18—discharge chamber, 19—screen, 20—quartz wedge, 21—light blocking screen, 22—diaphragm of 1.2 mm, 23—Videoscan VS-285C CCD camera, 24—photodiode, 25—Tektronix TDS2024b oscilloscope, 26—BNC575 pulse/delay generator, 27—thyratron driver PB-3DV.

**Figure 3 polymers-14-03940-f003:**
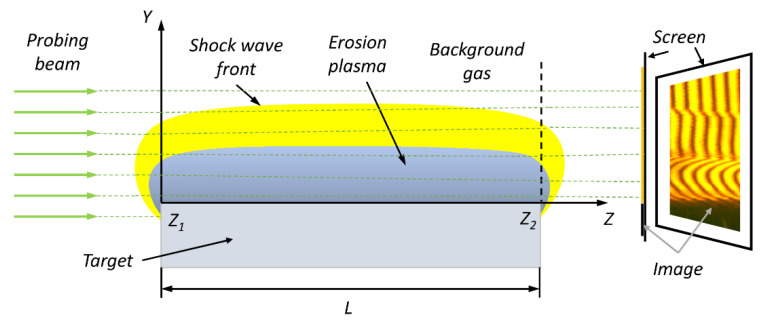
The probing beam passing through the inhomogeneity.

**Figure 4 polymers-14-03940-f004:**
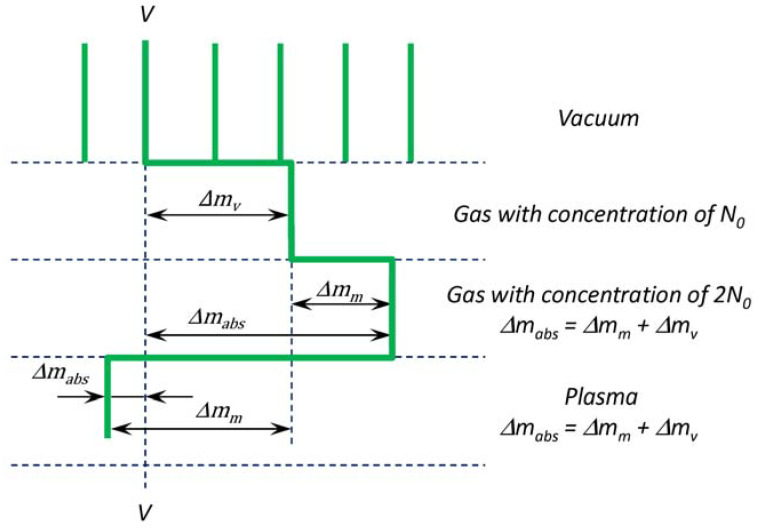
Hypothetical trend of the interference fringes paths in different situations: in vacuum, in gas with various concentrations, and in plasma.

**Figure 5 polymers-14-03940-f005:**
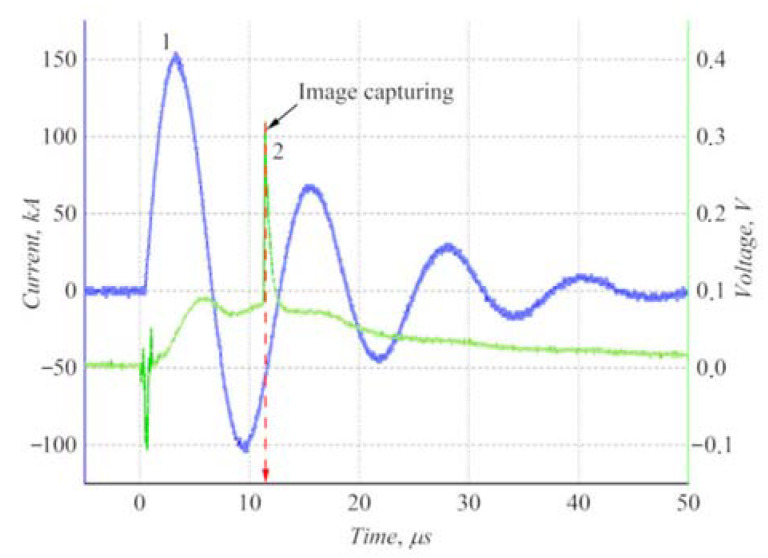
Typical current waveform (1) and photodiode signal (2). The peak corresponded to image capturing.

**Figure 6 polymers-14-03940-f006:**
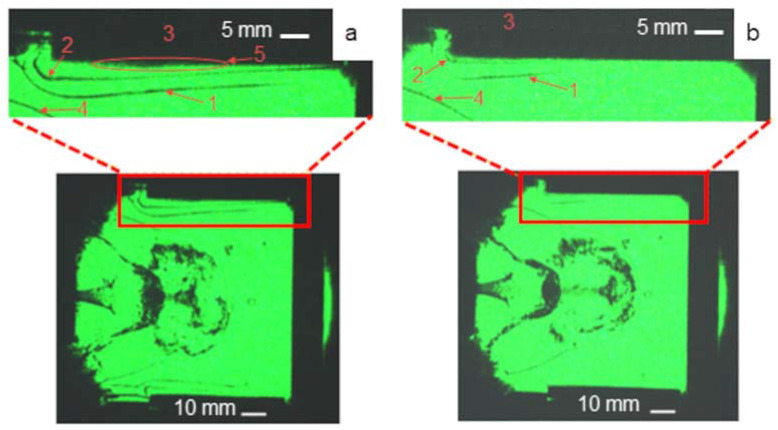
Schlieren images (at ≈11 μs after switching) of MPC discharge and gas-dynamic structures above the sample PTFE surface for pure neon (**a**) and neon/air mixture (**b**): 1—shock wave front from PTFE, 2—contact boundary, 3—PTFE sample, 4—shock wave front from MPC and 5—dense vapors.

**Figure 7 polymers-14-03940-f007:**
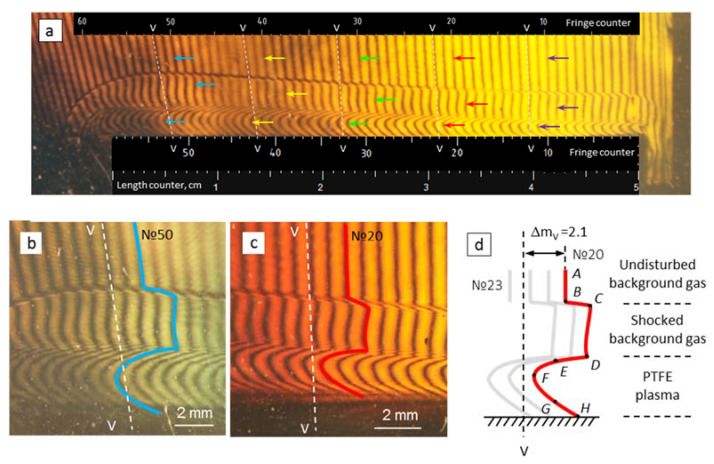
Interferogram of gas-dynamic response (**a**) with detailed fringe trends for #50 (**b**), #20 (**c**) and reconstructed response structure for #20 (**d**).

**Figure 8 polymers-14-03940-f008:**
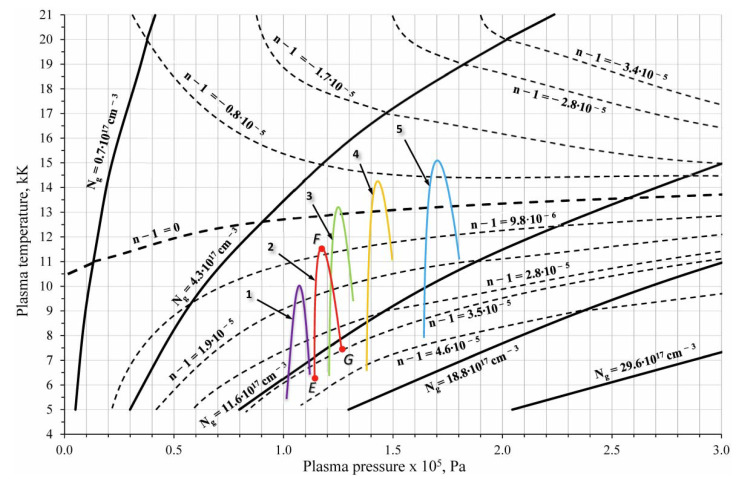
Parameters in the plasma for fringes # 10 (1), 20 (2), 30 (3), 40 (4) and 50 (5).

**Figure 9 polymers-14-03940-f009:**
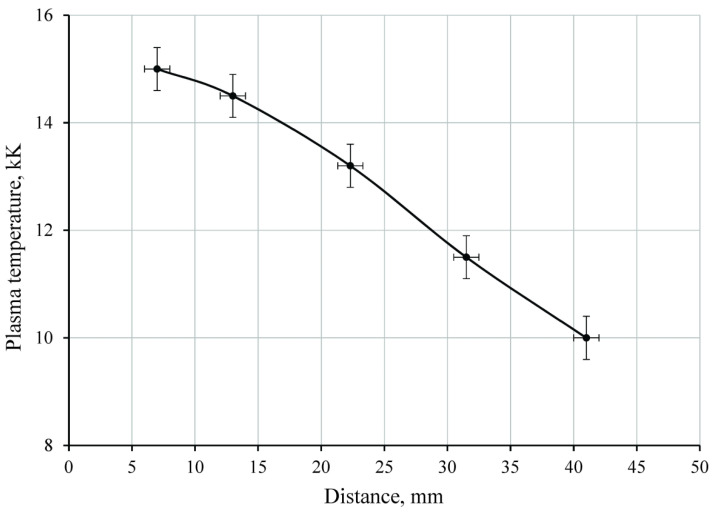
Distributions of the maximum plasma temperature along the sample length (for point F).

## Data Availability

The data presented in this study are available on request from the corresponding author.

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
