# Peer review of "Gas Dynamics Processes above the Polymers Surface under Irradiation with Broadband High-Brightness Radiation in the Vacuum Ultraviolet Spectrum Region"

_polymers, 2022, doi:10.3390/polym14193940_

Round 1

Reviewer 1 Report

This manuscript describes the plasma physics and gas dynamics for plasmas above PTFE samples. Although this work can be quite interesting, authors should think about substantial revision and consider these issues:

1) This work is mainly focused on the physics of the process not on the chemistry, materials and polymers. Why you submitted this manuscript to the Polymers journal? Please think about the fact that your topic is quite far from the scope of the journal.

2) Why don't You provide the characterization of your samples after all experiments? Again, if this manuscript would be submitted to Physics/Plasmas/Vacuum journals, I would agree that your study may be solely based on the characterization of the process. However, for Polymers, you need to specify what are the advantages/novelty of this research for Polymers audience. I encourage to characterize the samples after testing and present SEM, FTIR and XPS.

3) The references 36 and 37 appears after Ref9. This is not appropriate. Please change numbering accordingly.

Author Response

Dear Colleague!

First of all, we would like to thank you for interest in the work.

Concerning you remarks we can note the following

  1.  

Polymeric materials (PTFE, POM and others) are widely used as propellants in a design of current and novel pulsed plasma thruster (PPT) due to a possibility of PPT miniaturization, operation simplification etc. Polymers are undergone by exposure of high brightness and broadband radiation fluxes (including the UV/VUV ranges), extreme heat and mechanical loads which cause their ablation, material vaporization and vapor ionization. We note that there are some features of polymer ablation compared with evaporation of ceramics, metals and other materials. This circumstance leads to a variety of description of light-induced polymer ablation [see for example doi: 10.1021/cr010426b, 10.1007/978-3-319-96845-2_8, 10.1002/9780470594179.ch14]. In the present paper we used the optical techniques for obtaining data on ablation processes above PTFE being under extreme radiation, heat and mechanical conditions.      

That’s why we assume that our paper is directly connected to the topics of the Special Issue "New Polymeric Materials for Extreme Environments" and the obtained results are novel and useful to a wide range of professionals in material science, aerospace engineering, extreme matter and others.

  1.  

We claim that the main aim of the study is not associated with modification of PTFE for attainment of certain properties. We researched some processes of its ablation under the extreme conditions achieved with our PPT. So characterization of the samples does not seem useful for evaluation operation parameters of PPT with condensed polymer propellants. We assume that all our experimental techniques are most suitable for solution of the presented fundamental problem of PTFE ablation under exposure of the VUV/UV and plasma fluxes.

But we agree that it is necessary to correct the Introduction for highlighting and explanation of our purposes and used techniques.

  1. Yes. It is an occasion. We will fix it.

Reviewer 2 Report

Review of the manuscript entitled ‘Gas dynamics processes above the polymers surface under irradiation with broadband high-brightness radiation in vacuum ultraviolet spectrum region

The aim of the present study was to design a technique for quantifying of the response from irradiated polymers and get new information on polymer ablation under ultra-extreme VUV radiation.

The manuscript is very well written, clear and the figures can help to understand the experiments and the results. The results are challenging and confirm the interest of the set-up. The manuscript deserves to be published. A simple comment can be done on the presented work: no uncertainty is given in the manuscript concerning the results.

Author Response

Dear Colleague!

First of all, we would like to thank you for interest in the work and you opinion.

Round 2

Reviewer 1 Report

Although I am not completely satisfied with the answers and rebuttals from the authors, the provided information perhaps, can justify that this manuscript can fit the particular Special Issue. 

Some minor issues remained. I have to highlight that the quality of the newly introduced text it is not satisfactory. E.g. line 51, it seems that the authors did not make a proof-reading and there many issues with English. I suggest to ask a native speaker to work on the proof-reading. Again some references are of a bad style.

Author Response

Dear Colleague!

We tried to fix you remarks regarding these minor issues.

Best regards, A. Skriabin

Round 3

Reviewer 1 Report

The authors have made significant improvements and now the manuscript can be accepted for publication.